# Identification of Long-Distance Transport Signal Molecules Associated with Plant Maturity in Tetraploid Cultivated Potatoes (*Solanum tuberosum* L.)

**DOI:** 10.3390/plants11131707

**Published:** 2022-06-28

**Authors:** Zhiming Hui, Jianfei Xu, Yinqiao Jian, Chunsong Bian, Shaoguang Duan, Jun Hu, Guangcun Li, Liping Jin

**Affiliations:** Institute of Vegetables and Flowers, Chinese Academy of Agricultural Sciences, Key Laboratory of Biology and Genetic Improvement of Tuber and Root Crop, Ministry of Agriculture and Rural Affairs, Beijing 100081, China; huizhiming77@163.com (Z.H.); xujianfei@caas.cn (J.X.); jianyinqiao@caas.cn (Y.J.); bianchunsong@caas.cn (C.B.); duanshaoguang@caas.cn (S.D.); hujun@caas.cn (J.H.)

**Keywords:** potato, maturity, long-distance transport mRNA, metabolite

## Abstract

Maturity is a key trait for breeders to identify potato cultivars suitable to grow in different latitudes. However, the molecular mechanism regulating maturity remains unclear. In this study, we performed a grafting experiment using the early-maturing cultivar Zhongshu 5 (Z5) and the late-maturing cultivar Zhongshu 18 (Z18) and found that abscisic acid (ABA) and salicylic acid (SA) positively regulate the early maturity of potato, while indole-3-acetic acid (IAA) negatively regulated early maturity. A total of 43 long-distance transport mRNAs are observed to be involved in early maturity, and 292 long-distance transport mRNAs involved in late maturity were identified using RNA sequencing. Specifically, *StMADS18*, *StSWEET10C*, and *StSWEET11* are detected to be candidate genes for their association with potato early maturity. Metabolomic data analysis shows a significant increase in phenolic acid and flavonoid contents increased in the scion of the early-maturing cultivar Z5, but a significant decrease in amino acid, phenolic acid, and alkaloid contents increased in the scion of the late-maturing cultivar Z18. This work reveals a significant association between the maturity of tetraploid cultivated potato and long-distance transport signal molecules and provides useful data for assessing the molecular mechanisms underlying the maturity of potato plants and for breeding early-maturing potato cultivars.

## 1. Introduction

Potatoes (*Solanum tuberosum* L.) belong to the Solanaceae family, originated in the Andean Mountain region of South America, and are now the third most important food crop in the world, after wheat and rice [1]. Maturity is a quantitative trait controlled by multiple genes and is an indicator for breeders to identify suitable potato cultivars in different latitudes. Maturity is closely related to the onset of leaf senescence and the whole growth period, as well as associated time points of tuberization and potato yield [2]. While multiple SNP markers or genes have been detected in association with maturity, including solcap_snp_c2_47609 [3], solcap_snp_c2_22986 [4], solcap_snp_c2_11605 [5] and the *StCDF1* gene, the physiological and genetic mechanism of plant maturity in commercial tetraploid potato cultivars remains far from being unclear.

Potato maturity is closely related to plant growth and development [6]. Multiple nutrients, including phytohormones, secondary metabolites, and long-distance transport mRNAs, regulate plant growth and development via long-distance transport mediated by the plant vascular system [7]. Roots and shoots have an important mutual interaction: a high photosynthetic rate ensures high root activity by supplying a sufficient amount of photosynthates to the roots, while high root activity ensures a high photosynthetic rate by supplying sufficient nutrients to the shoots [8]. Multiple endogenous factors and environmental cues can influence whole plant senescence, including plant hormones, nutrient deficiency, pathogen infection, and extreme temperatures [9]. Plant hormones are important long-distance transport signal molecules [10,11] and regulate plant growth and development. Auxins typically promote the growth of plant leaves and roots, and abscisic acid (ABA) and ethylene promote leaf senescence [12]. Auxins inhibit leaf senescence, and the IAA content declines in senescing leaves [13,14]. The contents of indole-3-acetic acid (IAA), gibberellin A3 (GA3), and cytokinin (CTK) decrease, and the ABA content increases in early-maturing cultivars at maturity [15]. Abscisic acid (ABA) strongly induces the onset of leaf senescence, and precocious leaf yellowing can be caused by higher concentrations of endogenous ABA due to the upregulated expression of ABA biosynthesis genes [16]. Salicylic acid (SA) is a well-defined inducer of leaf senescence, whose contents progressively increase during leaf senescence [17]. In *Arabidopsis*, salicylic acid (SA) regulates leaf senescence by inducing upregulated expression of the transcription factor WRKY75 [17], and jasmonic acid (JA) regulates leaf senescence through the MYC2-Dof2.1-CYC2 transcriptional feedback loop [18].

Plant secondary metabolites are also found to be involved in regulating a series of complex processes of biochemical response and play an important role in plant growth and development [19]. Grafting has been widely used to study the signal molecules involved in long-distance transport [20]. Signal molecules, such as plant hormones, long-distance transport mRNAs, and other small molecules, are transported among organs through the vasculars [21,22]. In potatoes, mRNAs, including *StBEL5*, *StBEL11*, *StBEL29*, and *StPOTH1*, have been reported to regulate the formation of potato tubers via long-distance transport [23,24,25]. However, the molecular mechanism of long-distance transport mRNAs in regulating maturity is far from being unclear.

In this study, we analyzed phytohormone contents, contents of several secondary metabolites, and expression levels of long-distance transport mRNAs in the grafted plants of early-maturing and late-maturing potato cultivars using a combination of RNA-seq analysis and metabonomic assay. We found that scion cultivar maturity plays a vital role in regulating potato plant maturity, that ABA and SA play positive roles in regulating early maturity, and that IAA negatively regulates early maturity. *StMADS18*, *StSWEET10C*, and *StSWEET11* could be candidate genes associated with early-maturity traits in potatoes.

## 2. Results

### 2.1. Effect of Grafting on Maturity

To determine how grafting affects maturity, we selected one early-maturing cultivar (Zhongshu 5) and one late-maturing cultivar (Zhongshu 18) to generate grafted plants. Leaves of the Z5/Z18 (Z5 was grafted onto Z18) grafted plants and Z5/Z5 (Z5 was grafted onto Z5) self-grafted plants were yellow at 80 days after emergence (Figure 1a,b). The Z18/Z5 (Z18 was grafted onto Z5) grafted plants and Z18/Z18 (Z18 was grafted onto Z18) self-grafted plants grew well and had green leaves and blooming flowers (Figure 1c,d). These results demonstrate that the maturity of the scion cultivars plays a major role in mediating the maturity of the grafted plants.

### 2.2. Grafting Affects Tuber Yield

The average tuber yield of each of the grafted plants is shown in Figure 2. Compared with the Z5/Z5 grafts, the Z18/Z5 grafts showed significantly higher tuber yield per plant. The tuber yield per plant of Z5/Z18 grafts decreased compared with the yield of the Z18/Z18 grafts (Table 1), while there were no significant differences in tuber skin color or tuber shape between the Z5 and Z18 self-grafted plants and the reciprocally grafted plants (Figure 2).

### 2.3. Grafting Affected Hormone Contents

We measured the contents of the stem hormones ABA, SA, and IAA in the rootstocks and scions. The contents of abscisic acid (ABA) and salicylic acid (SA) of Z5/Z18 and Z5/Z5 grafted plants were inflated when compared with those of Z18/Z18 self-grafted plants (Table 1). The indole-3-acetic acid (IAA) contents in the stems of the Z18/Z5 and Z18/Z18 grafted plants were significantly higher than in the Z5/Z18 and Z5/Z5 grafted plants (Table 2).

### 2.4. Identification of Long-Distance Transport mRNAs Associated with Maturity

Large-scale transcriptome analysis was performed to detect the differentially expressed genes (DEGs) in stems of grafts to explore long-distance transport mRNAs associated with maturity. A total of 314 genes were co-upregulated in Z5/Z18-S, Z5/Z18-R, and Z5/Z5 stems in comparison to Z18/Z18 stems (Appendix A), while 1553 genes were co-upregulated in Z18/Z5-S, Z18/Z5-R, and Z18/Z18 stems in comparison to Z5/Z5 stems (Appendix A).

Of the 314 DEGs related to early plant maturity [8], 43 were co-upregulated in Z5/Z18-S, Z5/Z18-R, and Z5/Z5 stems but co-downregulated in Z18/Z5-S, Z18/Z5-R, and Z18/Z18 stems, which were considered long-distance transport mRNAs associated with early-maturity characteristics in potatoes (Appendix A; Figure 3a). Of the 1553 DEGs, 292 DEGs were co-upregulated in Z18/Z5-S, Z18/Z5-R, and Z18/Z18 stems but co-downregulated in Z5/Z18-S, Z5/Z18-R, and Z5/Z5 stems (Appendix A, Figure 3b).

### 2.5. GO Enrichment and KEGG Pathway Analysis of the Detected Long-Distance Transport mRNAs

First, we performed GO enrichment analysis of 43 long-distance transport mRNAs detected for their associations with early maturity and identified the genes enriched in 16 biological processes, 9 cellular components, and 5 molecular functions. The primary biological process categories were “cellular process (19 mRNAs)”, “metabolic process (17 mRNAs)”, and “response to stimulus (12 mRNAs)”. In cellular component terms, mRNAs from the early-maturing cultivar Zhongshu5 were classified into “cell (26 mRNAs)”, “cell part (26 mRNAs)”, “organelle (17 mRNAs)”, “membrane (16 mRNAs)”, and “membrane part (16 mRNAs)”. The primary molecular function categories were “binding (19 mRNAs)”, “catalytic activity (13 mRNAs)”, “transporter activity (8 mRNAs)”, and “transcription regulator activity (7 mRNAs)” (Appendix A). We then performed a KEGG pathway enrichment analysis with 43 long-distance transport mRNAs, which are associated with potato early-maturity traits [8], to investigate the primary biochemical metabolic and/or signal transduction pathways. The analysis found that these mobile mRNAs were significantly enriched in the taurine and hypotaurine metabolism pathways (Appendix A).

The GO enrichment analysis was also performed with the 292 long-distance transport mRNAs whose expression is associated with late-maturity potato traits. These mRNAs were enriched in 21 biological processes, 10 cellular components, and 12 molecular functions. The primary biological process categories were “cellular process (159 mRNAs)”, “metabolic process (136 mRNAs)”, “cellular component organization or biogenesis (76 mRNAs)”, “response to stimulus (60 mRNAs)”, “biological regulation (55 mRNAs)”, and “regulation of biological process (52 mRNAs)”. In cellular component terms, these mobile mRNAs were classified into “cell (166 mRNAs)”, “cell part (166 mRNAs)”, “membrane (124 mRNAs)”, “membrane part (106 mRNAs)”, “organelle (103 mRNAs)”, and “organelle part (57 mRNAs)”. The primary molecular function terms of these mobile mRNAs were “catalytic activity (139 mRNAs)”, “binding (115 mRNAs)”, “transcription regulator activity (19 mRNAs)”, and “transporter activity (15 mRNAs)” (Appendix A). KEGG pathway enrichment analysis of 292 long-distance transport mRNAs was then performed to determine the main biochemical metabolic and signal transduction pathways, the results of which demonstrated that these mobile mRNAs were enriched in terms of phenylpropanoid biosynthesis, one carbon pool by folate, and glycan degradation (Appendix A).

Forty-three long-distance transport mRNAs were identified from the stems of the early-maturing potato cultivar Zhongshu 5, including *CBF* (C-repeat binding factor) transcriptional factors, *MADS* transcriptional factors, and sugar transporter, *SWEET*. *CBF* can promote leaf senescence and natural dormancy in fruit trees, improve plant resistance, and inhibit plant growth [26,27]. Meanwhile, *MADS* is associated with fruit ripening [28], which promotes the physiological maturation of tomato fruits [29], plant senescence, and natural dormancy on fruit trees [30,31]. For example, in the apple dwarf rootstocks, the peach *PpCBF1* gene was transferred into the apple dwarf rootstock M.26 variety, and the *PpCBF1* T166 transgenic apple line inhibited the growth of the T166 plant, upregulated the expression of the apple *MdDAM* gene, and promoted early senescence and dormancy [32,33]. The apple *MdDAM* gene is a MADS transcription factor that promotes the physiological maturation of the tomato fruit. In pear trees, the pear PpCBF2 transcriptional factor regulates the expression of *PpMADS13-1* [30,31]. Furthermore, the sugar transporter (SWEET) family plays crucial roles in leaf senescence, carbohydrate transportation, development, and environmental adaptability. Some *SWEETs*, such as *OsSWEET5* and *PbSWEET4*, can promote rice and pear leaf senescence, respectively [34,35].

### 2.6. Identification of Metabolites Associated with Maturity

Metabolomic analysis with the grafts was performed to detect the metabolites associated with the maturity phenotype. A total of 401 annotated metabolites were detected in the stems. Twenty metabolites were co-upregulated in the Z5/Z18-S, Z5/Z18-R, and Z5/Z5 stems compared with the Z18/Z18 stems (Appendix A), which were potentially related to early-maturity traits [8]. Twenty-seven metabolites were co-upregulated in Z18/Z5-S, Z18/Z5-R, and Z18/Z18 stems compared with Z5/Z5 stems (Appendix A), which were potentially related to late-maturity characteristics in potato [8].

### 2.7. KEGG Pathway Analysis of Metabolites Associated with Maturity

We annotated the metabolites using the KEGG database. Twenty differentially expressed metabolites related to early-maturity plants were enriched in terms of arginine and proline metabolism, vitamin B6 metabolism, glycerolipid metabolism, and glutathione metabolism (Appendix A), while 27 differential metabolites related to late-maturity plants were enriched in the following pathways: biosynthesis of secondary metabolism, tryptophan metabolism, phenylalanine tyrosine, and tryptophan biosynthesis, indole alkaloid biosynthesis, and aminocyl-tRNA biosynthesis (Appendix A).

### 2.8. Identification of Graft-Transmissible Metabolites Associated with Maturity

In this study, 4 of 20 differentially expressed metabolites related to early-maturity traits were found to be co-upregulated in Z5/Z18-S, Z5/Z18-R, and Z5/Z5 stems, and were found to be co-downregulated in Z18/Z5-S, Z18/Z5-R, and Z18/Z18 stems. These were considered long-distance transport metabolites associated with potato plant early-maturity traits [8] (Figure 4a; Table 3). Of the four early-maturity-related metabolites, 1-caffeoylquinic acid, protocatechuic acid-4-glucoside, and trihydroxycinnamoylquinic acid are phenolic acids, while myricetin-O-glucoside-rhamnoside is a flavonoid. Since these early-maturity-related metabolites were not annotated in the KEGG database, their function remains unclear.

Of the 27 differential metabolites related to late-maturity traits, 7 differential metabolites were co-upregulated in Z18/Z5-S, Z18/Z5-R, and Z18/Z18 stems, and co-downregulated in Z5/Z18-S, Z5/Z18-R, and Z5/Z5 stems. These were considered long-distance transport metabolites associated with potato plant late-maturity characteristics (Figure 4b; Table 4). Of the seven long-distance transport metabolites related to plant late-maturity traits, there were three amino-acids and derivatives, L-histidine, L-tryptophan, and DL-alanyl-DL-phenylalanine, two phenolic acids, sinapyl alcohol, and sinapinaldehyde, two alkaloids (-)-cotinine, and methoxy indole acetic acid. The physiological function has not yet been reported.

Therefore, how these mobile metabolites regulate early maturity in potatoes requires further study. Integrated transcriptome and metabolome analysis was an important method of increasing our understanding of the molecular mechanism of numerous plant characteristics [36]. In this study, we constructed a correlation network of long-distance transport mRNAs and metabolites in early-maturity cultivars based on transcriptome and metabolome analysis, which identified some key point genes.

### 2.9. Integrated Analysis of Long-Distance Transport mRNAs and Metabolites

In this study, various long-distance transport signal molecules were identified for their association with maturity traits of potato plants. A correlation network analysis of long-distance transport mRNAs and metabolites associated with early-maturing traits in potatoes was performed using bioinformatic software (Perl scripts + the igraph R package) and presented in Figure 5. For early-maturity traits, 18 long-distance transport mRNAs were related to 1-caffeoylquinic acid, six mobile mRNAs were related to protocatechuic acid-4-glucoside, and 3 mobile mRNAs were related to trihydroxycinnamoylquinic acid. For late-maturity traits, 292 long-distance transport mRNAs were correlated with seven metabolites (L-histidine, L-tryptophan, DL-alanyl-DL-phenylalanine, sinapyl alcohol, sinapinaldehyde, (-)-cotinine, and methoxy indole acetic acid).

## 3. Discussion

Plant grafting technology has been widely used to study scion–rootstock interactions, substance long-distance transport, and signaling transduction [20]. This study combined transcriptome and metabolome analysis to explore long-distance transport signal molecules in the grafting system of early-to-late maturity potato cultivars.

It has been reported that grafting can induce changes in plant phenotype through scion–rootstock interactions, such as improving stress resistance, enhancing quality, and increasing production [21,22,37]. In this study, one early-maturing potato cultivar (Zhongshu 5) and one late-maturing cultivated potato cultivar (Zhongshu 18) were successfully reciprocally grafted under normal growth conditions. Using the material created, we compared the plant maturity and tuber yield per plant of different grafts. Our results demonstrated that the maturity of the scion had an important effect on the maturity and tuber yield per plant of the grafts. The yield of the early-maturing cultivar was increased when a late-maturing cultivar was grafted onto an early-maturing cultivar. This study provides new insights into yield traits by identifying the mechanism of scion–rootstock interactions.

In this study, phytohormone content analysis demonstrated that the ABA and SA content of the early-maturing cultivar grafted plants was higher than that of late-maturing cultivar grafted plants before the physiological maturation of the plant. Meanwhile, the IAA content of the early-maturing cultivar grafted plants was significantly lower than that of late-maturing cultivar grafted plants before the physiological maturation of the plant. Therefore, relatively high levels of ABA and SA and relatively low IAA levels can result in the early physiological maturation of potato plants.

Tetraploid potato plant maturity is an important quantitative trait, which is controlled by many alleles or allele dosage and is related to tuberization. The *StCDF1* locus is a master regulator of potato maturity, which contains 12 unique alleles, and *StCDF1.1* is a very late-maturity allele; *StCDF1.2* and *StCDF1.3* are early-maturity alleles at temperate latitudes [2,38]. Although the early-maturing *StCDF1.2* and *StCDF1.3* alleles were introduced into the very late diploid potato and the short-day-dependent *Solanum tuberosum* group *andigenum* and the transformed plants showed accelerated signs of senescence and terminated their life cycles earlier than the nontransgenic controls [2], the two early-maturing alleles are introduced into the late commercial tetraploid potato cultivars (*Solanum tuberosum* group *tuberosum*); whether the transformed plants can show accelerated senescence and terminated their life cycles earlier than the nontransgenic controls is still unreported. Therefore, the major role of *StCDF1* alleles for plant maturity needs to be verified in the commercial tetraploid potato cultivars further. Furthermore, potato plant maturity is related to tuberization, and a few signals, including StSP3D and StSP6A, correlate with potato tuberization [39,40], but it is still unclear whether these tuberization signals positively or negatively influence potato plant maturity.

This study assessed 43 mobile mRNAs and found that the transcription levels of *StCBF1*, *StCBF2*, *StMADS18*, *StSWEET10C*, and *StSWEET11* were upregulated before the physiological maturation of early-maturing potato cultivars. Additionally, the transcription levels of *StCBF1*, *StCBF2*, and *StMADS18* genes were co-upregulated in the stems of the early-maturing cultivar Zhongshu 5. This indicates that the StCBF1, StCBF2 protein may bind to the 5′-upstream regions of the *StMADS18* gene to promote potato leaf senescence. The sugar transporter SWEET family genes, *StSWEET10C* and *StSWEET11*, potentially promote potato leaf senescence via sugar metabolism and the transport pathway. This suggests that the mRNAs of the *StSWEET10C*, *St SWEET11*, and *StMADA18* genes act as long-distance transport mRNAs transported from the scion to the rootstock to promote plant senescence. Furthermore, the early-maturing *StCDF1.2* and *StCDF1.3* genes have not been found in the 43 mobile mRNAs, which the *StCDF1* allele transcript does not potentially move over long distances. Therefore, although these potential candidate genes are related to early-maturity traits, their molecular mechanism requires further study.

Plant secondary metabolites can function as signal molecules that affect plant growth and development. Sugars, as signaling molecules, have been found to modulate root/leaf differentiation, fruit ripening, leaf senescence, and tuber formation [41,42,43]. Sugar signaling is an important regulator of source-sink-regulated senescence (SSRS), and an increased accumulation of trehalose-6-phosphate promotes leaf senescence in maize [44]. Glucose mainly functions in seedling growth, photosynthesis, and leaf senescence, and sucrose is more closely related to maturation, such as the process of flowering and the development of storage organs [43]. Phenolic acids are ubiquitous in all higher plant tissues and act as a signal molecule in certain symbiotic relationships, such as defense molecules against soil pests and pathogens. Phenolic acid compounds are primarily synthesized by the phenylpropanoid biosynthetic pathway, including cinnamic acid, caffeic acid, and protocatechuic acid, all of which inhibit plant growth [45,46]. Caffeoylquinic acid compounds protect plants against predation and infection by inhibiting plant growth [47]. Furthermore, flavonoids [48], flavonols [49], and kaemperol-3-O-rhamnoside-7-O-rhamnoside [50] act as signal molecules to inhibit plant growth by hindering polar IAA transport. In this study, there are three phenolic acids compounds and one flavonoid glycoside related to early-maturity traits in potatoes; however, sugar signaling molecules are not found. There are three phenolic acid compounds (1-caffeoylquinic acid, protocatechuic acid-4-glucoside, and trihydroxycinnamoylquinic acid) and one flavonoid glycoside (myricetin-O-glucoside-rhamnoside) related to early-maturity traits in potatoes, which could potentially inhibit their growth before the physiological maturation of early-maturing cultivars. The IAA content of early-maturing potato cultivars is significantly lower than that of late-maturing cultivars prior to plant physiological maturation, which could be related to myricetin-O-glucoside-rhamnoside. Of the phenolic acid compounds and flavonoid glycoside associated with early-maturity traits, none were annotated in the KEGG pathway, and no mobile mRNAs associated with the phenylpropanoid biosynthetic pathway were found (which were the first metabolites discovered in potatoes).

## 4. Materials and Methods

### 4.1. Plant Materials

Two tetraploid potato cultivars (Z5, *Solanum tuberosum* L. cv. Zhongshu 5 and Z18, *Solanum tuberosum* L. cv. Zhongshu 18) were used as grafting materials for mutual grafting and self-grafting. Zhongshu 5 is an early-maturing cultivar with a growth period of approximately 60 days, and Zhongshu 18 is a late-maturing cultivar with a growth period of approximately 99 days. Four grafting combinations were performed: (1) Z5/Z18, in which the early-maturing cultivar Z5 was grafted onto the late-maturing cultivar Z18; (2) Z5/Z5, in which the early-maturing cultivar Z5 was self-grafted; (3) Z18/Z5, in which the late-maturing cultivar Z18 was grafted onto the early-maturing cultivar Z5; and (4) Z18/Z18, in which the late-maturing cultivar Z18 was self-grafted.

### 4.2. Grafting Procedure

Grafting was performed using grafting clips and the cleft grafting method described by Lee et al. [51]. Microtubes were sown in a net house in 2019. Grafting was performed after emerging when the plants had 5–7 leaves and were 12–15 cm high. The grafting union was fixed with grafting clips, and each grafting combination was 20 plants. Approximately 7 days after grafting, the grafted plants were not shared with a sunshade net until the scion was well connected with the rootstock. The grafting trials were performed in the Chabei (CB) region, Hebei Province (41°15′ N,114°07′ E,1500 m a.s.l., China), with average minimum and maximum temperatures of 10.05 °C and 21.66 °C, respectively.

### 4.3. Sample Preparation

We selected six samples to analyze differential long-distance transport molecules after grafting. The four samples for Z5/Z18 and Z18/Z5 mutual grafting were Z5/Z18-S (Zhongshu 5 scion), Z5/Z18-R (Zhongshu 18 rootstock), Z18/Z5-S (Zhongshu 18 scion), and Z18/Z5-R (Zhongshu 5 rootstock), while the other two self-grafted control samples were Z5/Z5 and Z18/Z18. Sampling from six separate plants was performed 15 days before the physiological maturation of the Z5/Z18 and Z5/Z5 grafted plants, at which point the stem tissue samples were collected (Figure 6). The sampling position was approximately 5 cm from the graft union. All target tissues were flash-frozen in liquid nitrogen and stored at −80 °C for further use. Three independent biological replicates were analyzed for phytohormone analysis, high-throughput sequencing, and wide-target metabolite sampling.

### 4.4. Identification of Plant Maturity and Tuber Yield of Grafted Plants

Growth period (GP) [52] was defined as the period between seedling emergence and physiological maturation (50% of plant leaves showing a yellow coloration). The emergence of individual seedlings was assessed every 5 days, starting 20 days after planting. Physiological maturation of the grafted plants was assessed every 5 days, starting 40 days after emergence. Tubers per plant were harvested after the grafted plants were killed by frost.

### 4.5. Phytohormone Analysis

The stem tissues of Z5/Z18-S, Z5/Z18-R, Z18/Z5-S, Z18/Z5-R, Z5/Z5, and Z18/Z18 were harvested 15 days before the Z5/Z18 and Z5/5 grafted plants matured. The samples were weighed and frozen in liquid nitrogen. IAA, ABA, GA, JA, SA, and ACC measurements were performed using an HPLC-ESI-MS/MS system (HPLC, Shim-pack UFLC SHI-MADZU CBM30A system, www.shimadzu.com.cn/ (accessed on 6 December 2019); MS, Applied Biosystems 6500 Triple Quadrupole, www.appliedbiosystems.com.cn/ (accessed on 6 December 2019) at Wuhan Metware Biotechnology Co., Ltd. The mean hormone contents of the samples were compared using a two-tailed Student’s *t*-test with pooled variance.

### 4.6. RNAseq Sample and Library Preparation

The stem tissues of Z5/Z18-S, Z5/Z18-R, Z18/Z5-S, Z18/Z5-R, Z5/Z5, and Z18/Z18 were harvested 15 days before the Z5/Z18 and Z5/5 grafted plants matured. Approximately two grafted plants from each graft combination were pooled to obtain one replicate, and data were collected from three biological replicates. RNA was extracted from the samples using a Spin Column Plant total RNA Purification Kit according to the manufacturer’s instructions (Sangon Biotech, Shanghai, China). The purity of the extracted RNAs was assessed on 1% agarose gels as well as by a NanoPhotometer spectrophotometer (IMPLEN, Los Angeles, CA, USA). For RNA quantification, we used a Qubit RNA Assay Kit in Qubit 2.0 Fluorometer (Life Technologies, Carlsbad, CA, USA). RNA integrity was assessed using the RNA Nano 6000 Assay Kit of the Agilent Bioanalyzer 2100 system (Agilent Technologies, Santa Clara, CA, USA). The construction of Illumina sequencing libraries was performed using previously described protocols [53]. The cDNA libraries were sequenced on the Illumina HiSeq platform (Illumina Inc., San Diego, CA, USA) by Wuhan MetWare Biotechnology Co., Ltd. (www.metware.cn (accessed on 18 December 2019), Wuhan, China).

### 4.7. Sequencing Data Analysis

The raw data (raw reads) were processed by removing adapter sequences. Clean reads were obtained by removing reads containing adapter and poly-N sequences and low-quality reads with low Q-values. The clean reads were mapped to the *Solanum tuberosum* genome using TopHat v2.0.12. The gene expression levels were determined based on the million base pairs sequenced (FPKM) values. Differential expression analysis comparing two treatments was performed using the DEGSeq R package (1.20.0). A corrected *p*-value of 0.005 and a log2(fold change) value of 1 was used as thresholds to identify differentially expressed genes [54,55].

### 4.8. Widely Targeted Metabolome Sample Preparation and Analysis

All procedures related to sample preparation, metabolome profiling, and data analysis were performed at Wuhan MetWare Biotechnology Co., Ltd. (www.metware.cn (accessed on 24 December 2019) following their standard procedures. The freeze-dried stem tissue samples of Z5/Z18-S, Z5/Z18-R, Z18/Z5-S, Z18/Z5-R, Z5/Z5, and Z18/Z18 were crushed to a powder using a MM 400, Retsch grinder. After weighing 100 mg of the crushed powder, aliquots were extracted at 4 °C with 0.6 mL of 70% aqueous methanol. To achieve a higher extraction rate, the aliquots were vortexed six times during the extraction process. The aliquots were then centrifuged at 10,000× *g* for 10 min to obtain a supernatant, after which the samples were filtered using a microporous membrane (0.22 µm) and further processed/stored for UPLC-MS/MS analysis.

### 4.9. Chromatographic Mass Spectrometry Acquisition Conditions

The data acquisition instrument system included ultra-performance liquid chromatography (UPLC) (Shim-pack UFLC SHIMADZU CBM30A, https://www.shimadzu.com.cn/, Tokyo, Japan (accessed on 24 December 2019) and tandem mass spectrometry (MS/MS) (Applied Biosystems 4500 QTRAP, https://www.appliedbiosystems.com.cn/ (accessed on 24 December 2019). The liquid phase conditions included column: (1) waters ACQUITY UPLC HSS T3 C18 1.8 µm, 2.1 mm × 100 mm; (2) mobile phase: phase A = ultrapure water (0.04% acetic acid was added), phase B = acetonitrile (0.04% acetic acid was added); (3) elution gradient: 0.00 min B = 5% in comparison, B was linearly increased to 95% in 10.00 min and maintained at 95% 1 min, 11.00–11.10 min, B was reduced to 5%, and was 5% balanced to 14 min; (4) flow rate 0.35 mL/min; column temperature 40 °C; injection volume 4 µL. The mass spectrometry conditions were as follows: the electrospray ionization (ESI) temperature was 550 °C, the mass spectrometry voltage was 5500 V, the curtain gas (CUR) was 30 psi, and the collision-induced dissociation (CAD) parameter was set to high. In the triple quadrupole (QQQ), each ion pair was detected based on optimized decolusting potential (DP) and collision energy (CE).

The material was characterized according to the secondary spectrum information based on the self-built database MWDB (Metware database) at Wuhan MetWare Biotechnology Co., Ltd. (www.metware.cn (accessed on 24 December 2019). The isotope signal was removed during the analysis, and the repeated signals, including K + ions, Na + ions, NH_4_ + ions, and fragment ions (which are other, larger molecular weight substances), were removed.

Metabolite quantification was performed using multiple reaction monitoring (MRM) in triple quadrupole mass spectrometry. In the MRM mode, the fourth-stage rod first screens the precursor ions (parent ions) of the target substance and excludes the ions corresponding to other molecular weight substances to initially eliminate the interference; the precursor ions break through the collision chamber to induce ionization and form fragment ions. The triple quadrupole filter was then used to select a desired feature fragment ion to eliminate nontarget ion interference, which made the quantification more accurate and repeatable. After obtaining metabolite mass spectrometry data for different samples, peak area integration was performed on the mass spectral peaks of all the substances, and the mass spectral peaks of the same metabolite in different samples were integrated [56].

### 4.10. Metabolomics Data Analysis

Data matrices with the intensity of metabolite features under FOP and control conditions were uploaded to the Analyst 1.6.1 software (AB SCIEX, Ontario, ON, Canada). For statistical analysis, missing values were assumed to be below the limits of detection, and these values were imputed with a minimum compound value. The relative abundance of each metabolite was log-transformed before analysis to meet normality. A Dunnett’s test was used to compare the abundance of each metabolite between the control and FOP. The false discovery rate was used for controlling multiple testing. The supervised multivariate method, partial least squares discriminant analysis (PLS-DA), was used to maximize the difference in metabolomes between the control and the FOP-treated samples. The relative importance of each metabolite to the PLS-DA model was checked using a parameter called the variable importance in projection (VIP). Metabolites with VIP > 1.0 were considered differential metabolites for group discrimination. Principal component analysis (PCA), hierarchical cluster analysis (HCA), GO enrichment, and KEGG pathway analysis were all performed using R software (www.r-project.org (accessed on 24 December 2019).

### 4.11. Identification Standards for Long-Distance Transport Signals Associated with Potato Plant Maturity

According to a root–shoot interaction hypothesis for high productivity of field crops, roots and shoots have an important mutual interaction: a high photosynthetic rate ensures high root activity by supplying a sufficient amount of photosynthates to the roots, while high root activity ensures a high photosynthetic rate by supplying sufficient nutrients to the shoots [8]. Co-upregulated differential genes or metabolites in the stem tissues of Z5/Z18-S, Z5/Z18-R, and Z5/Z5 grafted plants were considered long-distance transport molecules associated with early-maturity traits, while co-upregulated differential genes or metabolites in the stem tissues of Z18/Z5-S, Z18/Z5-R, and Z18/Z18 grafted plants were considered long-distance transport molecules associated with late-maturity traits.

## 5. Conclusions

In summary, this study detected a few dozen long-distance transport signal molecules associated with potato plant maturity. In addition, mobile mRNAs of *StCBF1*, *StCFB2*, and *StMADS18* were found to be co-upregulated in the stem tissues of grafted plants using the early-maturing cultivar as scion before physiological maturation. *StCBF1*, *StCFB2*, and *StMADS18* are vitally important genes in regulating the early-maturity potato traits. Additionally, these long-distance transport signal molecules associated with potato plant maturity, including phytohormones, mobile mRNAs, and metabolites, and the data presented in this study may be useful for understanding the molecular mechanisms underlying plant maturity in cultivated potato.

## Figures and Tables

**Figure 1 plants-11-01707-f001:**
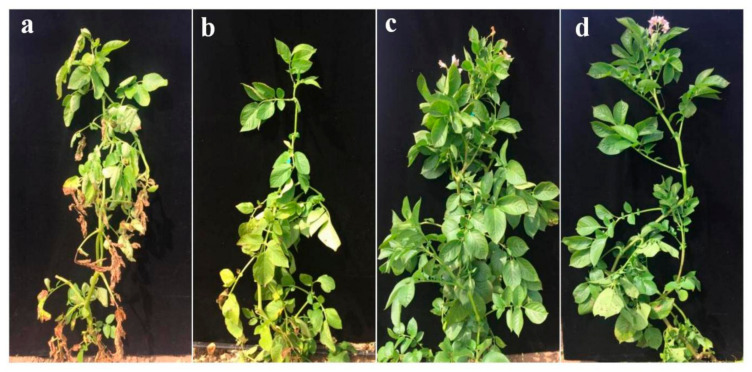
The grafted plant maturity at 80 days after emergence: (**a**) Z5/Z18 grafted plant, early-maturing cultivar Z5 was grafted onto late-maturing cultivar Z18; (**b**) Z5/Z5 grafted plant, early-maturing cultivar Z5 was self-grafted; (**c**) Z18/Z5 grafted plant, late-maturing cultivar Z18 was grafted onto early-maturing cultivar Z5; (**d**) Z18/Z18 grafted plant, late-maturing cultivar Z18 was self-grafted.

**Figure 2 plants-11-01707-f002:**
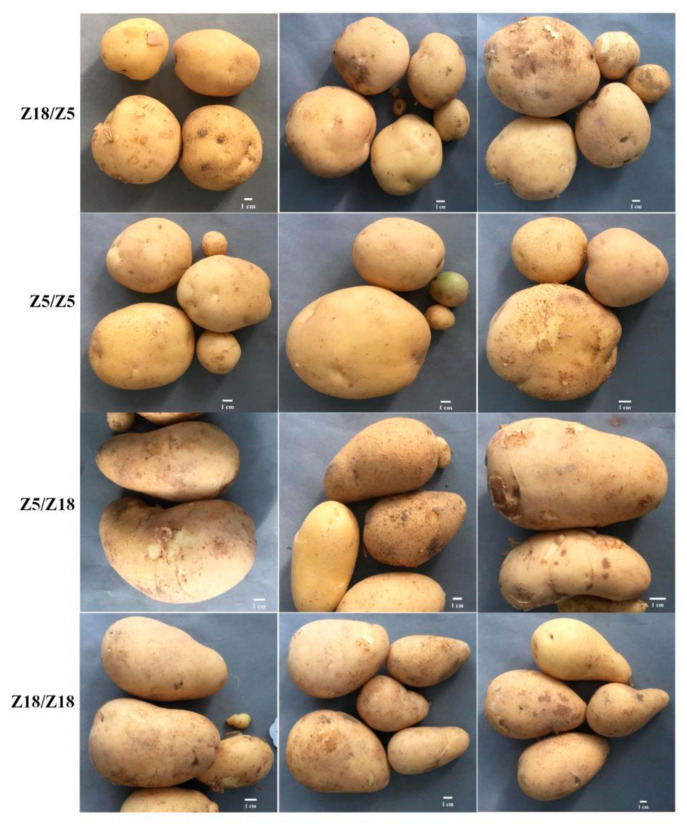
The tuber shape of four grafted combinations: tuber shape of Z18/Z5 and Z5/Z5 grafted plants is round; tuber shape of Z5/Z18 and Z18/Z18 grafted plants is long oval. Bar length is 1 cm.

**Figure 3 plants-11-01707-f003:**
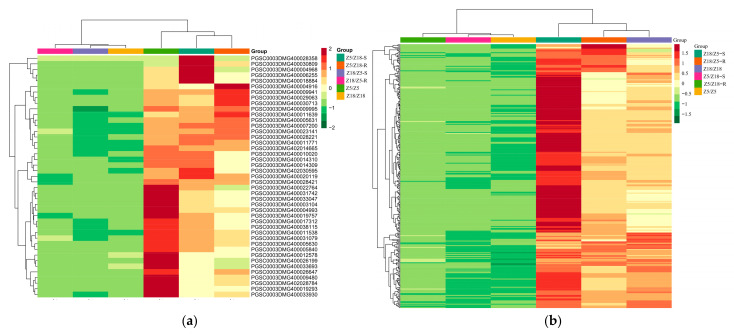
Clustering heat map of long-distance transport mRNAs related to early-maturity (**a**) and late-maturity (**b**) Z5/Z18-S: Z5 stems of Z5/Z18 grafts. Z5/Z18-R, Z18 stems of Z5/Z18 grafts. Z18/Z5-S: Z18 stems of Z18/Z5 grafts. Z18/Z5-R: Z5 stems of Z18/Z5 grafts. Z5/Z5: stems of Z5/Z5 grafts. 18/Z18: stems of Z18/Z18 grafts. Gene expression level by color: red indicates upregulated, green represents downregulated.

**Figure 4 plants-11-01707-f004:**
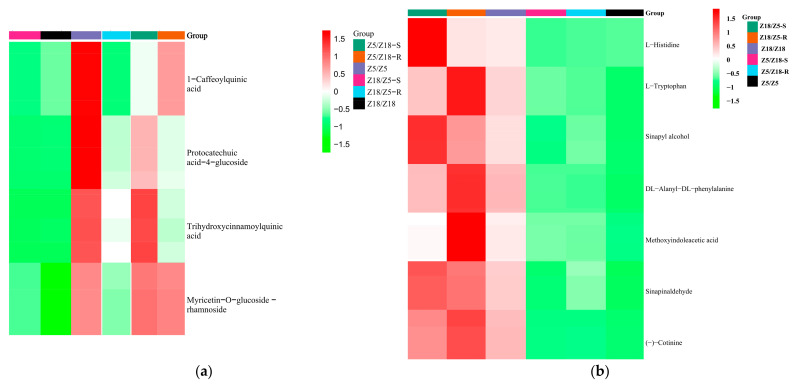
Clustered heat maps of long-distance transport metabolites associated with early-maturity (**a**) and late-maturity (**b**) traits in potato plants: (**a**) long-distance transport metabolite associated with early-maturity potatoes; (**b**) long-distance transport metabolites associated with late-maturity potatoes. Z5/Z18-S: Z5 stems of Z5/Z18 grafts. Z5/Z18-R: Z18 stems of Z5/Z18 grafts. Z18/Z5-S: Z18 stems of Z18/Z5 grafts. Z18/Z5-R: Z5 stems of Z18/Z5 grafts. Z5/Z5: stems of Z5/Z5 grafts. 18/Z18: stems of Z18/Z18 grafts. Metabolites accumulated level by color: red indicates up, green represents down.

**Figure 5 plants-11-01707-f005:**
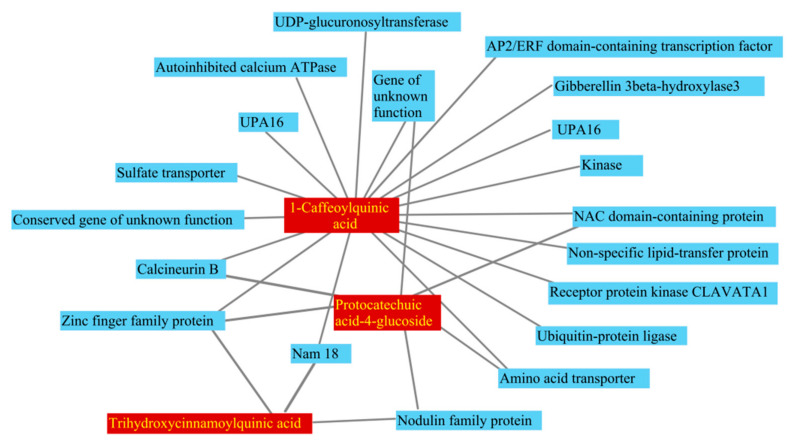
Network of long-distance transport mRNAs and metabolites related to early maturity: the coefficients of determination between long-distance transport mRNAs and metabolites are greater than 0.8.

**Figure 6 plants-11-01707-f006:**
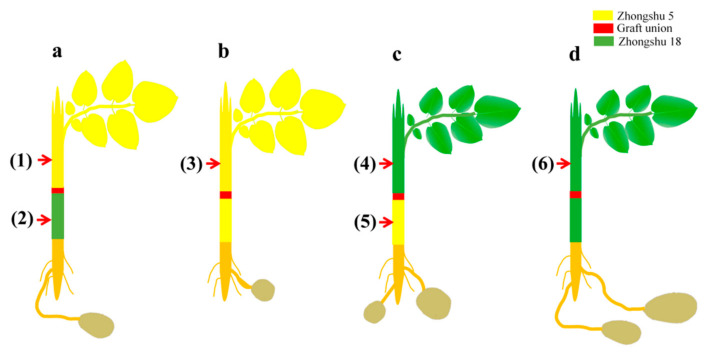
The sampling diagram of Zhongshu5 and Zhongshu18 mutual grafted and self-grafted plants: (**a**) Z5/Z18, early-maturing cultivar Z5 was grafted onto late-maturing cultivar Z18; (**b**) Z5/Z5, early-maturing cultivar Z5 was self-grafted; (**c**) Z18/Z5, late-maturing cultivar Z18 was grafted onto early-maturing cultivar Z5; (**d**) Z18/Z18, late-maturing cultivar Z18 was self-grafted. (1) and (2) were Z5/Z18-S (Zhongshu5 scion) and Z5/Z18-R (Zhongshu18 rootstock) stems, respectively; (3) was Z5/Z5 stems; (4) and (5) were Z18/Z5-S (Zhongshu18 scion) and Z18/Z5-R (Zhongshu5 rootstock) stems, respectively; (6) was Z18/Z18 stems. The red arrow indicates the sampling position at 5 cm from the graft union.

**Table 1 plants-11-01707-t001:** Analysis of mean tuber yield per plant of four grafted combinations.

Graft Combinations	Mean Tuber Yield Per Plant (g)
Z18/Z5	501.25 ± 300.50 ^a^
Z5/Z5	278.57 ± 108.54 ^b^
Z5/Z18	384.44 ± 197.87 ^ab^
Z18/Z18	555.00 ± 139.96 ^a^

Values are mean ± SD in six plants. Different letters in superscript represent significant (*p* ≤ 0.05) differences among mean yields in Z18/Z5, Z5/Z5, Z5/Z18, and Z18/Z18.

**Table 2 plants-11-01707-t002:** Abscisic acid, salicylic acid, and indole-3-acetic acid contents by HPLC.

Stem Tissues	ABA (ng/g FW)	SA (ng/g FW)	IAA (ng/g FW)
Z5/Z18-S	1.77 ± 0.78 ^a^	205.00 ± 137.45 ^ab^	10.40 ± 1.93 ^a^
Z5/Z18-R	1.39 ± 0.30 ^a^	109.23 ± 28.43 ^ab^	11.86 ± 3.32 ^a^
Z5/Z5	1.83 ± 0.33 ^a^	250.13 ± 167.13 ^a^	7.42 ± 3.04 ^a^
Z18/Z5-S	1.74 ± 0.16 ^a^	131.57 ± 40.34 ^ab^	27.70 ± 7.12 ^b^
Z18/Z5-R	1.23 ± 0.25 ^a^	87.30 ± 38.36 ^b^	23.13 ± 7.71 ^b^
Z18/Z18	1.18 ± 0.46 ^a^	139.00 ± 19.00 ^ab^	21.73 ± 5.37 ^b^

Values are mean ± SD in triplicate determinations. Different letters in superscript represent significant (*p* ≤ 0.05) differences among stems in Z5/Z18-S, Z5/Z18-R, Z5/Z5, Z18/Z5-S, Z18/Z5-R, and Z18/Z18: ABA, abscisic acid, SA, salicylic acid; IAA, indole-3-acetic acid.

**Table 3 plants-11-01707-t003:** KEGG pathway of four mobile metabolites related to early-maturity traits in potato plants.

Metabolites	Class	The KEGG Pathway
1-Caffeoylquinic acid	Phenolic acids	N/A
Protocatechuic acid-4-glucoside	Phenolic acids	N/A
Trihydroxycinnamoylquinic acid	Phenolic acids	N/A
Myricetin-O-glucoside-rhamnoside	Flavonoids	N/A

N/A: Information not available.

**Table 4 plants-11-01707-t004:** KEGG pathway of seven mobile metabolites related to the late-maturity traits in potato plants.

Metabolites	Class	The KEGG Pathway
L-histidine	Amino-acids and derivatives	Histidine metabolism
L-tryptophan	Amino-acids and derivatives	Tryptophan metabolism
DL-alanyl-DL-phenylalanine	Amino-acids and derivatives	N/A
Sinapyl alcohol	Phenolic acids	Phenylpropanoid biosynthesis
Sinapinaldehyde	Phenolic acids	N/A
(-)-Cotinine	Alkaloids	N/A
Methoxy indole acetic acid	Alkaloids	Tryptophan metabolism

N/A: Information not available.

## Data Availability

Data are contained within the article and Appendix A.

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
