# Peer review of "Identification of Long-Distance Transport Signal Molecules Associated with Plant Maturity in Tetraploid Cultivated Potatoes (Solanum tuberosum L.)"

_plants, 2022, doi:10.3390/plants11131707_

Round 1

Reviewer 1 Report

General Comments: This is a well-written manuscript on potato tuber maturity in tetraploid cultivated potatoes. As the authors mentioned, the molecular mechanism of plant maturity is not clearly understood. In commercial potato production plants, maturity is directly linked with crop productivity. The current study is well planned and executed. Authors have presented convincing data to show the association of cultivar maturity with long-distance transported signal molecules. The study enhances our understanding of the molecular mechanism of plant maturity, which could be very helpful in potato breeding programs. The data presented support the findings of the study.

The manuscript certainly has merits for publication. Therefore, I strongly support the manuscript for publication.

Good luck.

Reviewer 2 Report

This manuscript presents no more than the study the interaction between a scion and a rootstock after grafting two potato cultivars.  Title of manuscript should be edited accordance to results.

The maturity status of potato is defined on haulm senescence (haulm maturity), skin set (physical maturity), dry matter content (physiological maturity) and contents of sucrose, glucose and fructose (chemical maturity). Authors has defined physiological maturation of the grafted plants based on the length of period between seedling emergence and 50% of plant leaves showing a yellow coloration. That is correct only for leaves senescence detection. 

A period between seedling emergence and appearance yellow coloration of the early-maturing cultivar grafted onto late-maturing one and self-grafted plants was 80 days that is intermediate length for growth periods of two cultivars. So far as the early- maturing cultivar has a growth period of approximately 60 days and the late-maturing cultivar has a growth period of approximately 99 days, the intermediate length of period between grafted seedling emergence and their leaves senescence resulted from interaction between a scion and a rootstock.

Likewise, phytohormones content, metabolome profiling and transcriptome analysis are presented a momentary exactly reaction of two potato genotypes for grafting procedures.

 That is why Long-Distance Transport Signal Molecules identified in this study are associated not plant maturity. Their appearing was caused by a change in physiological processes after graftings.

Some additional comments are noted in the file

Reviewer 3 Report

I was excited to read this paper.  Maturity is an essential trait for potato breeders and I've been following the genetics work on cdf1 (the maturity locus). In general I think the grafting, metabolomics, and RNA-seq experiments were well done and interesting. However, I would have found this paper more useful if it linked more strongly to the genetics work.  Are any of these genes found related to the cdf1 locus?  I don't think this requires additional analysis, I think just a paragraph in the discussion would help.

I think line 84-86 is wrong.  I think it's supposed to read "Z18/Z5 grafted plants and Z18/Z18 self-grafted plants" not "Z18/Z5 grafted plants but Z18/Z18 self-grafted plants".

My final concern is that the standard deviations for several traits like yield and concentration of Salicylic acid are so big.  I think it would be good to provide some explanation for the amount of variation seen.
